# CD-IMM: The Benefits of Domain-based Mixture Models in Bayesian Continual Learning

**Daniele Castellana**
University of Florence
daniele.castellana@unifi.it

**Antonio Carta**
University of Pisa
antonio.carta@unipi.it

**Davide Bacciu**
University of Pisa
davide.bacciu@unipi.it

## Abstract

Real-world streams of data are characterised by the continuous occurrence of new and old classes, possibly on novel domains. Bayesian non-parametric mixture models provide a natural solution for continual learning due to their ability to create new components on the fly when new data are observed. However, popular class-based and time-based mixtures are often tested on simplified streams (e.g. class-incremental), where shortcuts can be exploited to infer drifts. We hypothesise that *domain-based mixtures are more effective on natural streams*. Our proposed method, the CD-IMM, exemplifies this approach by learning an infinite mixture of domains for each class. We experiment on a natural scenario with a mix of class repetitions and novel domains to validate our hypothesis.

## 1 Introduction

Continual learning (CL) is the ability to learn from a non-stationary stream of data. CL is fundamental in many real-life systems that are subject to concept drift, and whenever new data is collected over time [24]. The main challenge of continual learning is the stability-plasticity tradeoff, that is the tradeoff between the stability of the old knowledge and the plasticity necessary to learn from new data [14].

Recent results in the literature suggest that generative models may be more robust than discriminative models [34, 23, 16]. In this paper, we argue that one of the main benefits of generative models is their ability to factorize the data distribution as a mixture of smaller distributions. Mixtures allow to protect old components unrelated to new data from changes, while inferred probability of new data can be used

to detect drifts and create new components. However, popular solutions in the literature often exploit class labels [34, 16] or assume strong time coherence [23] to detect drifts (Section 2.4), simple mechanisms that may fail in most realistic settings. For example, in an object detection task, it is hard to assume that an object (or its appearance, such as the color) will never appear again (as in class-incremental or domain-incremental scenario, see Figure 1,2).

In this paper, we hypothesize that existing class-based or time-based generative methods will fail in a simple setting where new domains are discovered over time and classes are revisited (**H1-H3** in Section 3.2).

We propose the Class-Domain Infinite Mixture Model (CD-IMM), a new domain-based generative model which we expect to be more robust to class repetitions and novel domains (**H4**). CD-IMM lies on the Dirichlet Process Mixture Model (DPMM), a non-parametric model that adapts over time to the complexity of the data by adding more clusters when necessary. The method is general and it can work in online [27], task-free [2], or even unsupervised settings [30]. Furthermore, we expect that the clusters found by the CD-IMM will have a better correspondence with the natural clusters in the data distributions (**H4.1**).

We will test our assumptions on three different benchmarks. First, we propose Incremental Moons benchmark, a toy domain-incremental scenario based on the popular Moons dataset that we designed to showcase the advantages of Bayesian non-parametric classifiers and their robustness to domain drifts (**H1**). Then, we will use two different image classification benchmarks: Alphabet-Omniglot, where the class target for each character is its alphabet, and CIFAR100-Superclasses. The CD-IMM and the baseline methods will be tested on a simple scenario with class repetitions (**H1-H3**). Both datasets provide natural clusters (characters for Omniglot, classes for CIFAR100) that will be used to verify whether the clusters learned by the CD-IMM correspond to the natural clusters (**H4.1**).

Accepted pre-registered proposal at the 1[st] ContinualAI Unconference, 2023, Virtual. Full report to follow. Copyright 2023 by the author(s).

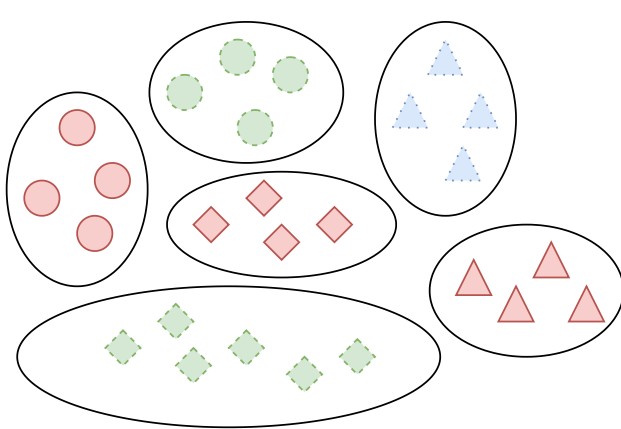

Figure 1: In real-world data, different classes (colors) may be unbalanced and structured into subgroups (shapes). We hypothesize that learning the latent structure of the probability distribution is helpful for continual learning.

## 2 Background and Related Works

### 2.1 Generative Models

Let us consider a learning task where we would like to learn a distribution $f(s)$ from a set of samples $\mathcal{D} = \{s_1, \ldots, s_N\}$ which are independently drawn from an unknown true distribution $p(s)$. The goal of generative models is to learn an approximated distribution $f(s, \theta)$ of $p(s)$, where $\theta$ are the model parameters which are adapted during the training.

For the purpose of our work, we focus on a specific type of generative model: the mixture model.

**Finite Mixture Models**   assume that the data distribution can be decomposed into a set of simpler distributions called components [28]:

$$f(s, \theta) = \sum_{z=1}^{K} p(s \mid z, \theta_z) p(z \mid \beta). \qquad (1)$$

Each component $p(s \mid z, \theta_z)$ has a set of parameters $\theta_z$, while the prior distribution $p(z)$ is parameterized by the vector $\beta$; both $\beta$ and $\theta_z$ are learned from data by using the Expectation-Maximisation (EM) procedure [10].

The generative process for the sample $s_i$ can be sketched as follows:

$$z_i \mid \beta \sim \text{Cat}(\beta), \qquad s_i \mid z_i, (\theta_c)_{c=1}^{K} \sim F(\theta_{z_i}). \qquad (2)$$

At first, the random variable $z_i$ is sampled using the categorical distribution $p(z) = \text{Cat}(\beta)$. The value $z_i$ indicates which component we should use to obtain the data $s_i$. Hence, $s_i$ is generated by sampling the distribution

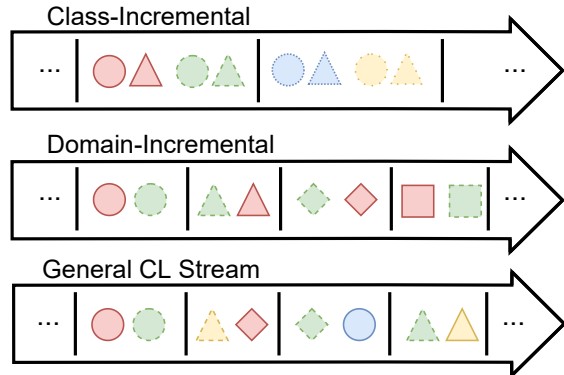

Figure 2: Bayesian CL methods exploit the structure of toy benchmarks (first two arrows), such as the lack of repetitions of classes and domains, to implicitly detect distribution drifts. We hypothesize that these methods will fail in real-world settings (last arrow) where new and old classes (or domains) are continuously encountered.

$p(s \mid z_i, \theta_{z_i}) = F(\theta_{z_i})$; $F$ is the family of the component distributions (e.g., a Gaussian) and $\theta$ are its parameters (e.g., mean and variance for the Gaussian).

A common drawback in mixture models is the choice of the number of components $K$. In fact, since the true distribution is unknown, it is hard to determine in how many components it can be decomposed

**Infinite Mixture Models**   overcome this limitation by allowing to adapt the number of components directly on the observed data. They are also known as Dirichlet Process Mixture Models (DPMM) [3] since they are based on Dirichlet Process (DP) [12].

For our purpose, it is convenient to define the DPMM generative process using the "stick-breaking" construction [32]:

$$\begin{aligned} \beta \mid \alpha &\sim \text{Stick}(\alpha) & z_i \mid \beta &\sim \text{Cat}(\beta) \\ \theta \mid H &\sim H, & s_i \mid z_i, (\theta_c)_{c=1}^{\infty} &\sim F(\theta_{z_i}), \end{aligned} \qquad (3)$$

where $\text{Stick}(\alpha)$ and $H$ are the prior of $p(z)$ and $p(s \mid z, \theta_z)$, respectively. Thanks to the prior specification, we can sample new components on the fly during the training, making the model infinite (i.e. $z_i \in [1, \infty]$). The generative process of each sample is equal to the finite mixture case: at first, a component $z_i$ is selected; then, the selected component is used to sample the data.

Due to the infinite capacity of the model, the posterior becomes intractable and the training cannot be performed with the EM algorithm anymore. Typically, approximated strategies are employed such as sampling [29] and variational inference [5]. It is worth highlighting that during the training the number of components $K$ is always finite.

## 2.2 Continual Learning

A continual learning stream is a sequence of datasets $\mathcal{D}_1, \ldots, \mathcal{D}_T$, where $\mathcal{D}_t = \{s_i \mid s_i \sim p_t(s)\}$ and $p_t(s)$ may change over time. While our method also works in unsupervised settings, we focus on supervised problems where $s = \langle x, y \rangle$, i.e. both the input $x \in \mathbb{R}^D$, and the class $y \in [1, C]$ are observed. Most methods in deep continual learning assume *virtual drift* [15], which means that the underlying data distribution, which we call the *real distribution* of samples $p(x, y)$, exists and is constant over time. Therefore, at each step, only a subset of this distribution is available for training. For example, in a *class-incremental* setting, we have $p_t(x, y) = p(x, y \mid y \in \mathcal{Y}_t)$, where $\mathcal{Y}_t$ is the set of classes visible at time $t$. Instead, in a *domain-incremental* setting $p_t(x, y) = p(x, y \mid z \in \mathcal{Z}_t)$, where $\mathcal{Z}_t$ is the set of subdomains visible at time $t$. Note that while class labels are often visible during training, the domain $z$ is a hidden variable.

## 2.3 Continual Mixture Models

In the continual learning setting, the objective of generative models is to learn a parametric approximation $f(x, y \mid \theta)$ of the joint distribution $p(x, y)$ from a stream of datasets $\mathcal{D}_1, \ldots, \mathcal{D}_T$ without storing old data. As stated before, the data $\mathcal{D}_t$ observed at time $t$ is obtained from a portion of the whole joint distribution, i.e. $\mathcal{D}_t \sim p_t(x, y)$.

**Time-based Mixture.** Since the joint distribution is discovered one portion at a time, it could be reasonable to partition the parametric approximation of the generative model. To this end, mixture models can be leveraged:

$$f(x, y \mid \theta) = \sum_t f_t(x, y \mid \theta_t) p(t), \qquad (4)$$

where $f_t(x, y \mid \theta_t)$ is the parametric approximation learned using the data $\mathcal{D}_t$. The variable $t$ represents the partition we are considering: when the distribution shift is known at training time, the variable $t$ is visible; otherwise, the value of $t$ is hidden.

The main advantage of this formulation is that we decompose the learning problem into $T$ independent sub-tasks: once we have learned the approximation $f_t$ for the partition $p_t(x, y)$, we do not have to change it anymore (i.e. $f_t$ is learned only on $D_t$). However, such a decomposition can be difficult to learn since, in general, each portion $p_t(x, y)$ can be as complex as the whole distribution $p(x, y)$.

**Class-based Mixture.** Another approach is to decompose the approximation $f$ according to the class label of the samples:

$$f(x, y \mid \theta) = \sum_y f_y(x, y \mid \theta_y) p(y), \qquad (5)$$

where $f_c(x, c \mid \theta_c)$ is the parametric approximation of the distribution of samples with class $c$, i.e. $p(x, y = c)$.

While this decomposition is reasonable from the task point of view, learning each approximation $f_c$ is difficult in the continual setting. In fact, to learn $f_c$ we need all the samples with class $c$, i.e. the set $\{(x, y) \mid y = c\}$. However, the elements in this set can be scattered into $D_1, \ldots, D_T$.

**Domain-based Mixture.** The last approach decomposes the approximation $f$ according to the domain of the samples:

$$f(x, y \mid \theta) = \sum_z f_z(x, y \mid \theta_z) p(z), \qquad (6)$$

where $f_z(x, z \mid \theta_z)$ is the parametric approximation of the distribution of samples with domain $z$.

While class and task information is usually known (if we focus on supervised tasks with visible task boundaries), the domain of a sample is usually unknown. This worsens the training procedure of domain-based mixture models since they should also estimate the sample domain. We believe that this is the main reason why, as far as we know, such mixture models have been not used in the literature.

## 2.4 Assumptions and Limitations of Generative Models for CL in the Literature

In a realistic setting, we do not expect to know when or how many times each domain occurs. In general, we expect to see both new domains and classes over time, possibly with repetitions. While this setup seems natural, methods in the literature make some restrictive assumptions about the types of drifts that are allowed.

**Simple input distribution:** some methods assume that there exists a pre-trained feature extractor that can be frozen and returns linearly separable features. For example, Deep SLDA [16] uses this assumption to model each class as a single Gaussian distribution.

**Knowledge about task boundaries** : more sophisticated methods either assume that drifts are known or that they are easily predictable. Many class-incremental methods, such as Ven, Li, and Tolias [34], use the presence of new class labels to determine drifts. This assumes that all the examples of a particular class are observed at the same time.

**No repetitions:** domains and classes are never repeated during training. More formally, many methods assume $\mathcal{Y}_t \cap \mathcal{Y}_{t'} = \emptyset, \forall t \neq t'$ and $\mathcal{Z}_t \cap \mathcal{Z}_{t'} = \emptyset, \forall t \neq t'$. Many methods that freeze previous components make this assumption [34, 31].

**Balancing:** data is balanced and uniformly complex. This assumption is used by architectural methods that cre-

ate separate components for each class or domain [34, 31].

These assumptions are difficult to satisfy with real-world data. Furthermore, methods exploiting these assumptions either fail to learn incrementally or become very inefficient.

We focus on three methods, which we briefly summarize below, to show the general strategies adopted by generative methods for continual learning scenarios.

**Simple input distribution: Deep SLDA [16]** is a class-based finite mixture model and it assumes that each class can be modelled as a Gaussian distribution learned by a linear discriminant analysis (LDA) classifier. Since the raw input space is not Gaussian in most nontrivial applications, Deep SLDA uses a frozen pre-trained feature extractor. Still, the Gaussianity is a strong assumption if we consider that the feature extractor is frozen and it was never trained on the data from the real distribution $p(\boldsymbol{x}, y)$. The advantage of this approach is that it doesn't suffer from forgetting since the online LDA algorithm is equivalent to offline training.

**Class-based Mixture: Class-VAE.** Ven, Li, and Tolias [34] proposes an approach, which we call Class-VAE in this paper, based on a class-based finite mixture model where each component is a Variational Auto-Encoder (VAE). In principle, the VAE can model any complex distribution, therefore it does not need to model explicitly each subdomain. A disadvantage is that training VAEs incrementally is still an unsolved problem [25]. Class-VAE avoids this limitation by restricting to a pure class-incremental setting without repetitions. The main limitation of the class-VAE is that it cannot be updated if new data for an old class becomes available.

**Time-based Mixture: CN-DPM.** Lee et al. [23] proposes a time-based infinite mixture model which trains a VAE and a classifier for each task. The VAE is trained to approximate the $p_t(\boldsymbol{x})$ of the true distribution, while the classifier models $p(y|\boldsymbol{x})$. A new VAE is initialized and trained whenever a new probability drift is automatically detected by computing the probability of the new data given the current model. In practice, CN-DPM has been tested only on class or domain incremental tasks, where drift detection is almost trivial. We hypothesize (**H3**) that CN-DPM will either create too many VAEs or fail to recognize drifts in a scenario with repetitions and multiple domains occurring at the same time.

## 2.5 Related Work

**Continual Learning Methods:** Bayesian methods provide a natural solution for continual learning. Popular methods such as EWC can be interpreted as Bayesian methods that exploit an approximation of the posterior to mitigate forgetting [19]. Building on the same intuition, IMM [22] merges the new and old model using the first two moments of the posterior distribution, approximated by a Gaussian. The idea of factorizing the model, possibly freezing old components, is exploited by architectural methods [31], which can also use the same components for task inference [1, 7] to remove the need for task labels. Recently, it was shown that self-supervised objectives are more robust to forgetting than supervised objectives [8, 13]. We expect Bayesian mixture models to behave similarly due to their natural ability to recognize domains, tasks, or classes.

**Fair Evaluation and Realistic Benchmarks:** Farquhar and Gal [11] show how seemingly minor details in the evaluation can affect the performance of CL models and argue for a fairer evaluation. In this paper, we follow the idea that a fair evaluation should be based on realistic assumptions on the stream distribution, such as dropping the constraints of no class repetitions [9, 17, 6]. It is often argued that class-incremental [33] scenarios are more difficult than domain-incremental ones. In this paper, we argue that class-incremental setting, as popularly used in the literature (no repetitions), trivializes many CL challenges such as drift detection, and allows for simple solutions such as freezing that do not generalize to more complex streams.

## 3 Hypotheses and Proposed Method

### 3.1 Our proposal: Class-Domain Infinite Mixture Model (CD-IMM)

The main novelty of our proposal is to explicitly consider the domains in the data generation process. Usually, different classes have different domains (Figure 1). Thus, we assume that each class is composed of subgroups (i.e. domains), which we model via the discrete latent variable $z_y \in [1, K_y]$. We use the subscript $y$ to emphasise that different classes have different domains; $K_y$ indicates the number of domains in the class $y$. We model the joint distribution as:

$$p(\boldsymbol{x}, y) = \sum_{z_y} p(\boldsymbol{x} \mid z_y, \boldsymbol{\theta}_{z_y}) p(z_y \mid y, \boldsymbol{\beta}_y) p(y), \quad (7)$$

where $\boldsymbol{x} \in \mathbb{R}^D$ can be the input data or features extracted from a frozen backbone.

The above decomposition is obtained by assuming that the data $\boldsymbol{x}$ is independent of the class $y$ given that we know the domain $z_y$. This conditional independence derives from our assumption that each class has different domains. Thus, if we know the domain $z_y$ we know also the class $y$. It is worth highlighting that this assumption is not a limitation since the domains are arbitrary: we can always split a

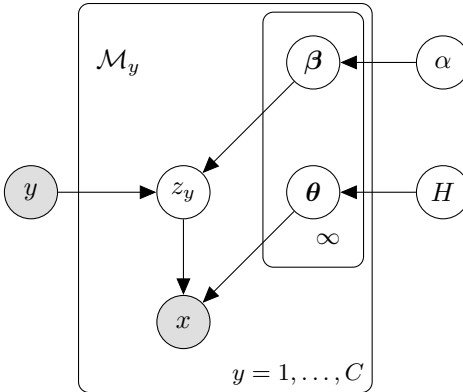

Figure 3: Graphical model of our proposal.

shared domain between two classes in two class-specified domains.

The generative model we propose in Equation 7 can be interpreted as a two-level mixture model. The first level is on the class labels: for each class $y$, we define a model $\mathcal{M}_y$ which is responsible for the generation of the data with label $y$. The second level is on the domains: each model $\mathcal{M}_y$ is again a mixture model which has a component for each domain $z_y \in [1, K_y]$. While the number of classes $C$ can be assumed to be known, the number of domains $K_y$ is usually unknown. Thus, we define each $M_y$ as a Dirichlet Process Mixture Model: for each class, the number of domains $K_y$ is theoretically infinite. The variables $\{\boldsymbol{\beta}, \boldsymbol{\theta}_1, \ldots, \boldsymbol{\theta}_\infty\}$ are the parameters of $\mathcal{M}_y$ where we removed the subscript $y$ to ease the notation. In Figure 3 we graphically represent our proposal.

**Definition of $\mathcal{M}_y$.** Each $M_y$ is modelled as a Gaussian DPMM [3]. Thanks to the DPMM, each $M_y$ can have an infinite number of components (i.e. domains). Each component $k$ is a multivariate Gaussian distribution with parameters $\boldsymbol{\theta_k} = \{\mu_k, \Sigma_k\}$, where $\mu_k \in \mathbb{R}^D$ is the mean and $\Sigma_k \in \mathbb{R}^{D \times D}$ is the covariance matrix. If needed, we can share the same variance across all the components obtaining a *tied* model as done in Deep-SLDA [16]. Also, we can constrain the covariance matrix to be diagonal, obtaining an isotropic multivariate Gaussian distribution. Due to the Bayesian fashion of DPMM, we must always define the prior $H$ of the parameters $\boldsymbol{\theta_k} = \{\mu_k, \Sigma_k\}$. To ease the computation, we define $H$ as a Gaussian-Inverse-Wishart distribution since it is the conjugate prior of $\boldsymbol{\theta_k}$. To be more precise, the prior is factorised as $H(\boldsymbol{\theta_k}) = p(\boldsymbol{\mu_k} \mid \Sigma_k)p(\Sigma_k)$ where: $p(\Sigma_k) = \mathcal{W}^{-1}(\boldsymbol{\Psi}, n_0)$ is the inverse Wishart distribution, and $p(\boldsymbol{\mu_k} \mid \Sigma_k) = \mathcal{N}(\boldsymbol{\mu_0}, \Sigma_k)$ is a multivariate Gaussian distribution with mean $\boldsymbol{\mu}_0$ and covariance $\Sigma_k$. Usually, $\boldsymbol{\mu}_0$ is the zero vector; however, this can lead to poor results (especially in the high-dimensional case) since the input data can be far from all the components that have means near zero. To overcome this issue,

we can initialise the mean of each component using the kmeans++ algorithm [4] using the first $B$ elements of the stream.

It is worth highlighting that our proposal can reduce to SLDA if $M_y$ is defined as a single Gaussian (i.e. each class has only one domain).

**Learning Procedure.** Since we observe the class labels, each model $\mathcal{M}_y$ is trained separately. Let us consider a new training sample $s_i = (\boldsymbol{x}_i, y_i)$, the input $\boldsymbol{x}_i$ is considered only to train the model $\mathcal{M}_{y_i}$.

The training of $M_y$ is based on the computation of the posterior $p(\boldsymbol{z}, \boldsymbol{\beta}, \boldsymbol{\theta})$, which is intractable due to the infinite number of clusters. We rely on a variational truncated approximation which defines a maximum number of components. While this might seem the same as a finite model, it is not [5]. The training can also be performed online (i.e. one update for each sample) by applying the Stochastic Variational Inference (SVI) framework [18].

### 3.2 Our Hypothesis

Our hypothesis is that domain-based mixture models are a better solution for more realistic streams with class and domain repetitions. The following hypothesis state how we expect class-based (**H1-H2**) and time-based method (**H3**) methods to fail, while domain-based mixture models learn the natural domains (**H4**).

**H1 - Simple class-based mixture models (Deep SLDA) fail to model complex multi-domain distributions.** Simple models such as SLDA do not work when data is not Gaussian.

**H2 - Complex class-based mixture models (Class-VAE) fail in settings with class repetitions.** Training the Class-VAE on a new domain for some old class will result in catastrophic forgetting.

**H3 - Time-based mixture models (CN-DPM) do not account for multiple domains appearing at the same time.** CN-DPM is unable to recognize small domain shifts or multiple domains appearing at the same time. (*H3.1*) Since the CN-DPM was tested on simple class-incremental settings, we expect CN-DPM drift detection to heavily rely on different class labels.

**H4 - Domain-based mixture models (CD-IMM) are able to learn on general streams with new classes and domains, even with repetitions** CD-IMM learns domains, independently from when they occur. (H4.1) The clusters discovered by our proposal match some intrinsic properties of the data.

# 4    Experimental Protocol

We experiment with three benchmarks in different configurations: Incremental Moons, Omniglot-Alphabets, CIFAR100-Superclasses. The chosen baselines are Deep SLDA, Class VAE, and CN-DPM, which provide representative methods for the class-based and time-based mixture models. We will test the batch setting, which allows for multiple epochs on each batch, and the online setting with a single pass on the data and without task boundaries.

**Domain-Incremental Moons.**    In order to provide an intuitive visualization of the CD-IMM, we propose Incremental Moons, a toy stream which is an incremental version of the Moons dataset[1].

Each moon is shaped as an arc of a circle plus some small Gaussian noise. Each moon is a different class, and the stream provides different sections of the arc for each moon, which are observed in a domain-incremental fashion. As a result, clusters do not overlap each other, and they are well separated. However, they are close enough to each other that it is not possible to approximate each cluster with a single Gaussian emission.

**Class-Incremental with Repetitions.**    To experiment with more realistic streams, we follow a setup similar to [17]. We split each class by domain and we randomly shuffle all the domains. Then, we group domains together to form a batch of data. As a result, new domains for already known classes can occur over time. To split classes into well-defined natural domains, we use datasets that provide coarse and fine labels. We use CIFAR100 [20] by using the 100 classes as (latent) domains and the 10 superclasses as classes, which we call CIFAR100-Superclasses. Similarly, we use Omniglot [21] with the alphabet as the target class and the character as the domain, which we call Omniglot-Alphabet. We will group class-domains randomly to obtain 10 drifts.

**Experimental Setup.**    Deep SLDA, Class VAE, and CD-IMM need a fixed backbone as a feature extractor. We will use raw features for Incremental Moons, a feature extractor pre-trained on CIFAR10 for CIFAR100 (as done in [34]), and a random MLP on Omniglot. If the random feature extractor for Omniglot is not powerful enough, causing all three methods to fail, we plan to document this failure and use a subset of the data to pre-train a feature extractor. The remaining hyperparameters ($\alpha, \mu_0, \Sigma_0$ for the CL-IMM) will be found via hyperparameter search evaluated on a separate validation stream for all the methods.

---

[1]an implementation of the Moons dataset can be found at https://scikit-learn.org/stable/modules/generated/sklearn.datasets.make_moons.html

**Metrics, Evaluation, Reproducibility.**    We will use the *accuracy over the whole target distribution*, estimated on a separate test set, as the main evaluation metric. Given $A_t$, the accuracy of the model at time $t$ on the entire test set, we will show the final accuracy $A_T$, average accuracy over time $\sum_{i=1}^{T} A_i$, and learning curves (**H1-H4**). We will also show a *confusion matrix* split by subdomain, which will allow us to check if time-based and class-based models are biased in the expected ways (**H2-H3**). For the CD-IMM, we will use a *domain confusion matrix*, where $DCM_i^j$ is the percentage of examples of data-domain $i$ associated with the model-domain $j$, to check how the natural domains fit within the model's domains (**H4**). Each method will be run 5 times to compute the mean and standard deviations for the metrics. We will release the source code for our experiments using Avalanche [26] to ensure the reproducibility of our results.

# 5    Experiments

## 5.1    Domain-Incremental Moons

In this experiment, we expect the following results:

- **H1** SLDA will fit each moon with a single Gaussian distribution, resulting in a low accuracy.

- **H2** Class-VAE will remember only the final domain (arc section) for each class due to catastrophic forgetting [25].

- **H3** There are two possible failure modes for CN-DPM: (1) it may not recognize new domains, resulting in catastrophic forgetting as the Class-VAE, and (2) it may not be able to learn the VAEs if too many drifts are detected, resulting in underfitted components.

## 5.2    Class-Incremental with Repetition

We expect the same failures detailed in Section 5.1.

## 5.3    Latent Domains Analysis

We expect CD-IMM to learn the natural subclasses in Alphabet-Omniglot and CIFAR100-Superclasses (**H4**). We don't expect the same natural clustering to occur with CN-DPM.

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
