# OpenReview forum: "CD-IMM: The Benefits of Domain-based Mixture Models in Bayesian Continual Learning"
_continualai.org/CLAI/2023/Unconference_Preregistration_Track — 1st CLAI Unconf_

### Official Review · Reviewer_sPAu · 2023-08-16
**Interesting idea**

**Clarity:** 3
**Originality:** 3
**Soundness:** 3
**Significance:** 3
**Rating:** 6
**Confidence:** 2

**Review:**

The paper presents a generative approach for continual learning based on domain-based Dirichlet Process Mixture Models (DPMM). In particular, the paper proposes to construct a DPMM for each class, enabling it to theoretically represent infinite domains for it.
The paper will test the proposed method on three different datasets: Incremental Moons, Alphabet-Omniglot, and CIFAR-100 Superclasses.

**Strengths:**

- The paper is well written and all the ideas are clearly explained.
- The paper is technically sound and proposes a promising approach to realistic continual learning settings.


**Weaknesses:**

- The paper proposed to learn a DPMM model for each separate class. This will increase the training time and the computational complexity required by the model. The paper should compare the computational complexity of this approach with other choices (both generative and discriminative).
- The set of baselines is limited to direct competitors of the proposed approach. However, nowadays the literature on continual learning is wide and it would be better to compare with multiple approaches taken from the state of the art.

**Questions:**

None

**Protocol:**

The paper presents a limited experimental protocol. It will report experiments on three small-scale datasets, not proving the general validity of the proposed approach. I would suggest adding a large-scale dataset to the benchmark (such as Imagenet) to prove the proposed method can be a viable solution for real-world use cases.

---

### Official Review · Reviewer_mcU8 · 2023-08-20
**Review for CD-IMM**

**Clarity:** 3
**Originality:** 3
**Soundness:** 2
**Significance:** 2
**Rating:** 6
**Confidence:** 4

**Review:**

The paper provides valuable insights in improving the generative classifier in continual leanring. The presentation is good but have space to improve; see the weakness below. The idea is novel. More comprehensive experiments are needed to make this work significant; see below weakness.

**Strengths:**

The paper propose a novel idea to formulate the mixture of models in generative classifier for continual learning. It propose to also consider the domains underlying the classes even in the class incremental tasks, which is insightful and can replicate some cases in the real world. The author also includes thorough analysis of previous related work, which provides valuable background to this work and can show the superior of the algorithm design.

**Weaknesses:**

-  Be careful about the terminology. Use something like generative model-based continual learning instead of continual generative methods or generative continual learning to avoid the misunderstanding of the goal of the paper.
- It is not necessary to repeated stated the limitations in existing CL methods with generative classifier. (both in related work and in section 3.2, though it is stated in the format of hypothesis in section 3.2)
- It is concerned that the improvement of the proposed methods is from increasing the parameter size but not from the modeling of domains, since it seems like the proposed method contains more Dirichlet Process Mixture Models, which practically contains more parameters to learn. It is suggested to use some techniques, such as ensemble, when compare with baseline methods
-  The paper is built on the motivation that "generative models may be more robust than discriminative models"[35].  [35] is a workshop paper that  only compare few old discriminative CL methods with generative classifiers, which makes the motivation not solid enough. It is suggested to either cite recent conference quality paper with comprehensive analyses or perform comparison of the proposed method with recent discriminative CL methods. Or rather remove this claim. Nevertheless, comparing the proposed method with recent popular discriminative CL methods and showing superior results would add merits to this paper.

**Questions:**

Please see the weakness above.

**Protocol:**

- For fairness in the experiment, see weakness above.
- The backbone structure and parameter size are not clear, which is important for the mixture modeling
- It is suggested to explore datasets with more distinct class domains, referring to those in [35]

---

### Official Review · Reviewer_yxCW · 2023-08-21
**Addressing Real-world problem of classes and domains evolution in data stream in continual learning, good paper, simple and incremental Bayesian Models**

**Clarity:** 3
**Originality:** 3
**Soundness:** 3
**Significance:** 3
**Rating:** 7
**Confidence:** 4

**Review:**

The paper is addressing a real-world problem of handling stream of data composed of new and old classes and domains over time. In doing so, they adopt Bayesian mixture models for modeling the class and domain variation as an  infinite mixture of domains (domain-based mixture models) for each class and by explicitly modeling domains in the data generative process. The proposed methods claim to better handle class repetitions and novel domains in data streams.

The problem statement is well motivated and it is worth investigation of real-world problem in handling data streams with evolving domains and classes.

**Strengths:**

+ paper is well formulated and clearly written
+ clear motivation however missing realistic examples to motivate the problem statement
+ idea is simple / intuitive and extend the baseline Bayesian mixture models
+ methodology is incremental
+ sound related works and methodology description
+ addressing the real-word problem of handling evolving domains and classes over time in data streams

**Weaknesses:**

- incremental (limited novelty)

Comments:
- page 1, column 2 and Para 3: This para is sudden pop-up without related context and referenced claims.
- Unclear realistic examples missing? Motivation by figures are clear however, realistic examples would bring life to the motivation to the problem statement, for instance, what the domains and class labels?
- Results and analysis section is expected to include the qualitative results on the forgetting over each time step for the historical data points in the stream of data and domain change
- Qualitative analysis of results is expected
- Experimental setup is unclear, why is the datasets chosen ?

Questions:
- how to handle a class label that has several domains? or a several domains share the same class label?
- how to handle heterogeneous data distribution over time as the assumption is Gaussian distribution for the input data stream?
- how would the approach/methodology applies to different data modality like text data?
- how does approximate inference handle infinite number of classes / domains ?

**Questions:**

- how to handle a class label that has several domains? or a several domains share the same class label?
- how to handle heterogeneous data distribution over time as the assumption is Gaussian distribution for the input data stream?
- how would the approach/methodology applies to different data modality like text data?
- how does approximate inference handle infinite number of classes / domains ?
- the choice of the data selection is unclear for the experimental setup
- how does the approach applies to different data modality for example to text? for topic modeling task?

**Protocol:**

- choice of data selection for experiments is unclear
- how does the approach applies to different data modality for example for text ?
- the applicability of the proposed approach to text data for example for topic modeling task would clearly highlight the contribution of the paper
- experiments must include the results/quantitative analysis on forgetting over time occurs for the historical data at each time step
- qualitative analysis of results is also expected, how does the model performs for certain data points comparing to the baseline models ?
- qualitative analysis for domains analysis is expected, why handling domains explicitly outperform the baseline models?
- it is unclear what are the domains ?
- please illustrate with examples the performance of the mode in cases where domains share the same class?

---

### Official Review · Reviewer_133x · 2023-08-22
**Review for Submission9**

**Clarity:** 3
**Originality:** 4
**Soundness:** 2
**Significance:** 4
**Rating:** 8
**Confidence:** 4

**Review:**

Goal: Learn components of the often-used Bayesian Mixture Models (BMMs) when the data stream involves the continuous occurrence of new classes and their subgroups (i.e. domains)
Problem: Existing practices around BMMs used for this goal are prone to distributional drifts in non-stationary settings.
Solution: Authors hypothesize that the shortcomings of the class-based and the time-based models warrant a new "domain-based" method they named CD-IMM.

The contributions of this paper are two-fold:

1) Demonstrate there exists a problem with the current methods
2) Propose a "domain-based" model

While the problem setting is very interesting and well-formulated, I am not sure about their proposed model. Especially, their conditional independence assumption below Equation 7. Would this not be equivalent to saying we have sum([K_{y} for all y]) classes?

Additionally, we still have that p(y) term in Equation 7. In other words, we still depend on the class.

For instance, what if there exists a latent domain that is inside both y_{a} and y_{b} for some a,b? This would violate their conditional independence assumption. Hence, they too assume there are "no repetitions" in the data stream.

Hence, I think that their proposed CD-IMM model seems like a class-based model in disguise. It is just a bit misleading to say that you are not class-based when you have that conditional independence assumption. Hence, a renaming should be done.

On a seperate note, their random feature extractor for Omniglot may suffer from information leakage.

Lastly, I found the introduction section a bit unclear:

1) “Recent results in the literature” refers to a single paper.
2) “Simple mechanisms that may fail in most realistic settings”. Why do they fail? Why domain-based models are superior to the class-based ones? The proposition that such models fail is part of the hypothesis along with the proposed CD-IMM solution.
3) CD-IMM abbreviation is undefined. CD-IMM model is also undefined. What differentiates it from a DPMM?

I recommend the introduction section to be revised in order to reflect that the problem formulation is also part of the hypothesis and that the novelties are presented in a concise manner.

I recommend this work to be accepted because of its relevance and ingenuity but the authors should clarify the above points raised.

**Strengths:**

Significant

Sufficiently sound

Clear yet slightly misleading

**Weaknesses:**

Conditional Independence Assumption

**Questions:**

Please elaborate on how different your model is from a class-based model with sum([K_{y} for all y]) classes

**Protocol:**

OK.

---

### Decision · Program_Chairs · 2023-09-12

**Decision:**

Accept

**Comment:**

Dear authors,

Congratulations, your paper has been accepted at the ContinualAI Unconference 2023! We look forward to engaging in further discussions with you and others in the community.

Details will follow shortly regarding camera-ready versions. Please do take the feedback from reviews into account.

Thanks!